# Assessing the stability of Pd-exchanged sites in zeolites with the aid of a high throughput quantum chemistry workflow

Hassan A. Aljama[1], Martin Head-Gordon [2,3✉] & Alexis T. Bell [1✉]

Cation exchanged-zeolites are functional materials with a wide range of applications from catalysis to sorbents. They present a challenge for computational studies using density functional theory due to the numerous possible active sites. From Al configuration, to placement of extra framework cation(s), to potentially different oxidation states of the cation, accounting for all these possibilities is not trivial. To make the number of calculations more tractable, most studies focus on a few active sites. We attempt to go beyond these limitations by implementing a workflow for a high throughput screening, designed to systematize the problem and exhaustively search for feasible active sites. We use Pd-exchanged CHA and BEA to illustrate the approach. After conducting thousands of explicit DFT calculations, we identify the sites most favorable for the Pd cation and discuss the results in detail. The high throughput screening identifies many energetically favorable sites that are non-trivial. Lastly, we employ these results to examine NO adsorption in Pd-exchanged CHA, which is a promising passive $NO_x$ adsorbent (PNA) during the cold start of automobiles. The results shed light on critical active sites for $NO_x$ capture that were not previously studied.

---

[1] Department of Chemical and Biomolecular Engineering, University of California, Berkeley, CA, USA. [2] Department of Chemistry, University of California, Berkeley, CA, USA. [3] Chemical Sciences Division, Lawrence Berkeley National Laboratory, Berkeley, CA, USA. ✉email: mhg@cchem.berkeley.edu; alexbell@berkeley.edu

Zeolites are microporous aluminosilicates composed of corner-sharing $TO_4$ tetrahedra (T = Si or Al). Substitution of trivalent Al for tetravalent Si creates a charge imbalance that is compensated by a proton or a metal cation. If the charge-compensating cation is a proton, the zeolite becomes a strong solid acid that is an effective catalyst for promoting a wide variety of hydrocarbon conversion reactions as well as the synthesis of a broad range of organic compounds. Metal cation-exchanged zeolites, where extra-framework metal ions replace some fraction of the protons, can also serve as adsorbents and catalysts[1–6].

In recent years, considerable insights into zeolite-catalyzed reactions have come from the application of quantum chemical analyses of the free energy and enthalpy landscapes governing the progress of chemical reactions[7,8]. A significant challenge for such studies is selection of the structure of the active site. To appreciate the issue, we must first recall that there are over 200 known zeolites structures, many of which have T sites occupied by Si and Al having symmetries differing from one another. Each of the charge-exchange sites is associated with a framework Al atom, but the distribution of Al is generally not known and is thought to be controlled by the kinetics of zeolite synthesis. A further complication is that the charge-exchange site involves an Al atom bonded to four oxygen atoms, any one of which can accommodate the proton. Further complexity arises when extra-framework metal cations replace protons. Monovalent cations are usually large enough that they bridge two framework oxygen atoms, while divalent cations can interact with two or three Al sites simultaneously.

It is, therefore, evident that the selection of a representative cation site is challenging and that full exploration of the chemistry on all possible sites is computationally formidable because of cost. For these reasons, many higher level theoretical studies of zeolite-catalyzed reactions[9–12] using density functional theory (DFT) have chosen to focus on a few active sites selected on the basis of limited experimental evidence and/or physical intuition. While this choice leads to a more tractable set of calculations, the downside is that potentially important active sites might be missed because they are difficult to identify experimentally or are not physically intuitive. Indeed, growing numbers of DFT studies have progressed from studying single T sites[13,14] to address greater complexity[15,16].

With this trend in mind, the objective of this work is to present and apply a DFT-based computational framework to identify the energetically most favorable adsorption sites (cation or proton) in any zeolite using a systematic high-throughput approach. The underlying calculations employ hybrid quantum mechanics/molecular mechanics (QM/MM) models[17,18] to capture extended environment around each active site, using high quality density functionals[19] to attain acceptable accuracy. Our approach starts by evaluating the possible zeolite structures that arise for different Si/Al ratios, then focuses on searching for energetically favorable cation/proton(s) adsorption sites for each structure using a lower level of theory (e.g., smaller basis set and cheaper functional) as a filter. The most favorable cation adsorption sites are then evaluated at a higher level of theory. We note that while there are previous high-throughput approaches in zeolites, they were largely focused on grand canonical Monte Carlo simulations[20–24].

To illustrate our approach, we use the challenging example of the siting of extra-framework Pd ions in Al-doped chabazite (CHA) and beta (BEA). Pd-CHA and Pd-BEA were chosen because they are good candidates for passive $NO_x$ adsorbers (PNAs) that can be used to trap the emissions of $NO_x$ from automobile exhaust during cold startup and before the three-way catalytic converter becomes effective[25–27]. The speciation of Pd in these materials is recognized as a challenging, and still controversial problem[28,29]. We limit our discussion to the following Pd species: $Pd^+$ associated with isolated charge-exchange sites, $Pd^+H^+$ pairs associated with two charge-exchange sites involving next nearest neighbor (NNN) or next-next nearest neighbor (NNNN) pairs of Al atoms, and $Pd^{+2}$ associated with two charge-exchange sites also involving NNN or NNNN Al pairs. After establishing a large set of feasible charge-exchange sites for Pd in CHA and BEA, we conclude by assessing the performance of the sites for NO adsorption on Pd-CHA, with some interesting results. Beyond this application, the goal is to provide a methodology that is transferable to other adsorption/catalysis problems in other zeolites. This can help shift the focus of DFT-based modeling of functional zeolite materials from a limited number of specific sites to a more systematic approach that allows more complete exploration of the descriptor space.

## Results and discussion

**High-throughput approach.** Figure 1 shows the workflow for the high-throughput approach to determine which charge-exchange sites are most favorable for accommodating the charge-compensating cation and/or proton. Briefly, Al atom(s) are first introduced at different tetrahedral sites in the bare $Si_xO_y$ zeolite, generating structures with unique (i.e., distinguishable) Al positions. For each unique Al arrangement, distinctions are made between atoms in the QM region, which are expected to be active in the adsorption process, and the surrounding atoms in the MM region. To determine the most energetically favorable location for accommodating the extra-framework cation, potential cation-exchanged sites are first enumerated and then surveyed by QM/MM calculations at the generalized gradient approximation (GGA) level of theory, followed by further evaluation of the top 5 most stable sites at the more refined (and computationally demanding) hybrid GGA (hGGA) level of theory. Lastly, results for all structures are compared to determine the structures where Al position(s) yield the most stable charge-exchange site. Each step is discussed in detail in the next few sections. This approached allowed us to perform over 7000 explicit DFT calculations in a systematic way, which is more than an order of magnitude higher than what is done typically in DFT studies of zeolites[13,14,16,30–32].

**Distinguishable Al locations.** The first step of the approach (Fig. 1) is determining the set of distinguishable Al sites in the zeolite structure. Univalent structures could be easily generated by replacing each unique Si T-atom in the zeolite by an Al atom. However, generating Al pairs is not trivial due to the 3D structure of zeolites. While this can be done manually for CHA, since it only has one unique T-site, it is more challenging to do for BEA, which has 9 T sites. Hence, we developed an automated method to generate all possible Al pairs. We focused in this work on pairs in the NNN (where the two framework Al are separated by a single Si atom) and the NNNN (where the two framework Al are separated by two Si atoms). Al pairs in nearest neighbor (NN) position were avoided in order to satisfy the Loewenstein rule[33]. Exceptions to Loewenstein's rule have been reported in the literature; however, these cases remain an exception and not the norm[34,35]. Pairs involving Al atoms farther apart than an NNNN configuration were not considered since as the Al–Al distance increases, the impact of having the Al pair diminishes and each atom starts behaving as an isolated Al site (some tests demonstrating this point are shown in Supplementary Figs. 1, 2). We note that this approach is easily applicable to any zeolite material and can be extended beyond NNNN pairs.

Our approach uses a molecular graph that identifies the connectivity of each atom in the cluster[36] (visually illustrated in Fig. 2). Each Al pair structure is generated by first identifying a

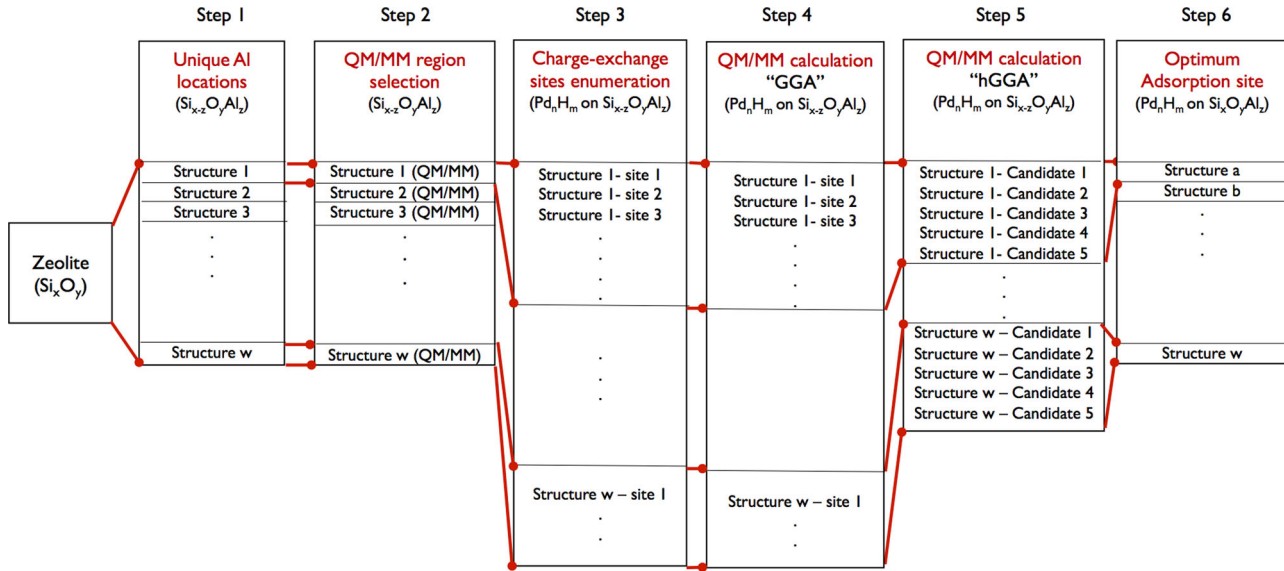

**Fig. 1 Workflow of the high-throughput approach used in this work.** $Pd_nH_m$ ($n = 0$, 1 or 2 and $m = 0$, 1, or 2) is used as an illustration. Starting with the zeolite material, structures are generated by identifying the unique Al position(s) in step 1 (generating structures from 1 to w), followed by identifying atoms in the QM and MM region for each structure in step 2. In step 3, possible charge-exchange sites are enumerated and then further evaluated in step 4 using QM/MM calculations at the GGA level. The 5 most favorable exchange sites from step 4 for each unique Al position are further evaluated at the higher hGGA level of theory in step 5. Finally, all structures from step 5 are ordered based on their energy stability, yielding the most energetically favorable sites for the exchange in the zeolite in step 6.

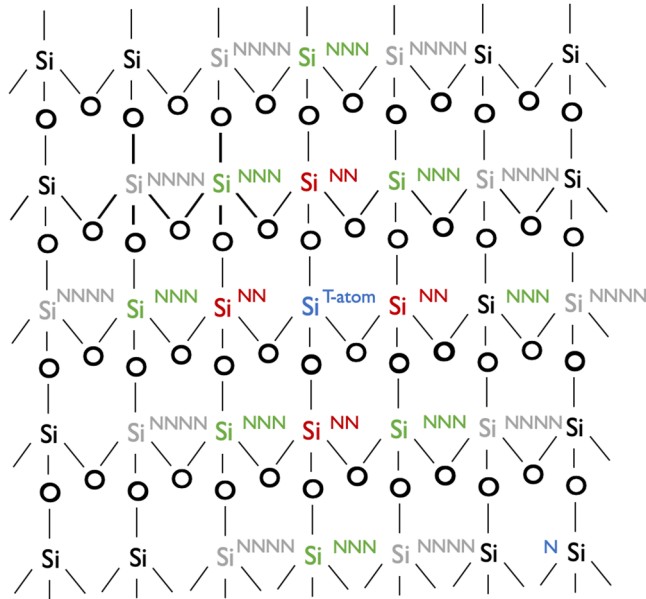

**Fig. 2 Schematic illustration of atomic positions relative to a tagged $Si^{T-atom}$ (in blue).** The $Si^{T-atom}$ is replaced with an Al atom to generate a univalent structure. For divalent structures, $Si^{NN}$ (in red), $Si^{NNN}$ (in green), and $Si^{NNNN}$ (in gray) atoms are identified. Candidate Al pair structures are generated by replacing one of the $Si^{NNN}$ or $Si^{NNNN}$ atoms, and the reference $Si^{T-atom}$ atom with Al atoms.

unique T-site ($Si^{T-atom}$). Based on the connectivity of the $Si^{T-atom}$, its NN Si atoms are identified ($Si^{NN}$). The Si atoms in NNN ($Si^{NNN}$) are identified by finding the next neighbors of $Si^{NN}$ (excluding the original $Si^{T-atom}$ and $Si^{NN}$). Each NNN Al pair is generated by replacing the $Si^{T-atom}$ and one of the $Si^{NNN}$ with Al atoms. For the NNNN Al pairs, the next neighbor of the $Si^{NNN}$ atoms are identified ($Si^{NNNN}$) (excluding $Si^{T-atom}$, $Si^{NN}$ and $Si^{NNN}$). Similarly, the $Si^{T-atom}$ and each one of the $Si^{NNNN}$ atoms

are then replaced with Al atoms to generate the NNNN Al pairs. Each structure with a unique Al location is identified by a unique index after the zeolite name (e.g., CHA-13) (full details on the xyz coordinates for the structures are available in the Supplementary Information).

Since the procedure described above can produce duplicate structures, we relied on calculating the nuclear repulsion energy (full results are available in the Supplementary Information). The method was verified using a BEA unit cell (which contains 36 T-atoms) and generating 36 different structures by replacing each Si T-atom with an Al atom. By using the nuclear repulsion energy, we were able to recover the 9 unique T sites. Overall, this results in 26 and 212 unique structures for CHA and BEA, respectively. In order to make the number of calculations more tractable, we further reduced the number of candidates by eliminating structures that share the same connectivity (i.e., the Al atoms in structures share the same types of $n$-membered-ring ($n$-MR)). Although in those cases structures are not exactly the same, they are structurally very similar (Supplementary Fig. 3). This reduced the final numbers to 100 structures for BEA and 12 structures for CHA.

**Selection of the QM and MM regions.** The second part of the workflow (Fig. 1) defines the atoms that comprise the QM region and the surrounding MM atoms in the cluster. Appropriate choice of the QM region is critical for the QM/MM calculations since the number of QM atoms must be sufficient to be accurate, but not too large in order to avoid significant computational cost. For structures with an isolated Al atom, the Si and O atoms that create a 4, 5 or 6MR with the Al atom are included in the QM atoms. If the structure contains an Al pair, and if the two pairs form a 4, 5, 6, 7, or 8MR, we include all Si and O atoms that are part of the $n$-MR. Also, for those structures, we include Si and O atoms that are part of a 4 or 5MR with either of the two Al atoms. Atoms comprising an $n$-MR are identified following the procedure described in Supplementary Fig. 4. The QM region includes on average 54 atoms, excluding hydrogen atoms used to

terminate the Si atoms. The QM regions used here are larger than used in previous work[12,31,37–39] and we found it to be more than sufficient for the calculations to be converged with respect to number of QM atoms (Supplementary Fig. 5). Finally, for the calculations of NO adsorption in Pd-CHA, the number of QM atoms was extended as needed to account for potential interactions of NO with other Si/O atoms in the framework.

**Enumeration of charge-exchange sites**. For each structure with a unique Al arrangement, we survey the energy landscape by placing the adsorbate(s) at multiple initial positions near each oxygen atom adjacent to the Al atom(s), as shown in Supplementary Fig. 6. For structures with an isolated Al atom, this requires only four calculations for either $Pd^+$ or $H^+$. For structures with an Al pair, finding the optimum position of a $Pd^{+2}$ requires eight calculations per structure, and finding the optimum position for $H^+H^+$ requires 16 total calculations. The most complicated case is for a $Pd^+H^+$ site near an Al pair, where in addition to surveying for $Pd^+$ charge-exchange site, a proton must be present near the opposite Al atom to compensate for the missing charge. This situation requires a total of 32 calculations per structure. We note that these numbers reflect the maximum number of calculations attempted. For some very unfavorable initial positions, the calculations did not converge. This is mostly limited to the GGA level search and happens in <10% of the total calculations (for the most part, this is due to placing Pd at the center of a 4MR). The extended number of initial positions in the scheme inevitably leads to some poor initial conditions. All generated structures and optimized geometries are available in the Supplementary Information.

**Survey of charge-exchange sites**. The major part of the computational cost in the workflow (Fig. 1) is associated with searching for the global minimum energy of the charge-exchange site, steps 4 and 5. For zeolites with multiple T sites, the number of calculations is significant. Carrying out all the calculations at the range-separated hybrid functional level of theory is intractable; however, it has been shown previously that this level of theory is needed to reach close agreement with experimental values[40]. Accordingly, we first use the B97-D3 exchange functional (which is at the GGA level) as a filter to determine the five most favorable exchange positions for the cation/proton(s) per each unique Al arrangement. For those five candidates, further calculations are done at the range-separated hybrid level using the $\omega$B97X-D exchange functional to determine the most energetically favorable position of the cations per each unique Al arrangement. We tested this approach on a number of structures by comparing the results between doing the full calculations using only $\omega$B97X-D to the approach described earlier. We found this approach to yield virtually the same results with significant reduction in computational cost (Supplementary Table 1).

In order to compare the stability of the Pd cation at different cation-exchanged sites, we calculated the energy of reaction ($\Delta E_{form}$) for the following two reactions:

$$Pd_{(g)} + H^+Z^- + \frac{1}{4}O_{2(g)} \rightarrow Pd^+Z^- + \frac{1}{2}H_2O_{(g)} \qquad (1)$$

$$Pd_{(g)} + H^+H^+Z^{-2} + \frac{x}{4}O_{2(g)} \rightarrow Pd^{+x}H^+_{2-x}Z^{-2} + \frac{x}{2}H_2O_{(g)} \qquad (2)$$

Equation 1 is used for an isolated Al zeolite and Eq. 2 for structures with an Al pair. $x$ is the oxidation state of Pd (either 1 or 2), $Pd_{(g)}$ is a gas phase Pd atom, $H^+$ is the compensating proton, Z is the charged zeolite framework, $H_2O_{(g)}$ and $O_{2(g)}$ are water and oxygen in the gas phase, and $Pd^{+x}$ is the Pd adsorbed

in the zeolite framework. Equations 1, 2 allow comparing the relative stability of sites with different Al configurations and oxidation states by using a consistent reference. The equations also rely on using a Brønsted site as a reference. This eliminates the impact of the thermodynamics of Al placement in the zeolite (which is kinetically driven during the synthesis of the zeolite[41]). For NO adsorption on Pd-exchanged CHA, we use the following equation to calculate the NO adsorption energy ($\Delta E_{NO}$):

$$\Delta E_{NO} = E_{Pd*NO*} - E_{Pd*} - E_{NO_{(g)}} \qquad (3)$$

where $E_{Pd*NO*}$ is the total energy of NO adsorbed on the Pd-exchanged zeolite in the DFT calculation, $E_{Pd*}$ is the total energy of the Pd adsorbed on the zeolite framework and $E_{NO_{(g)}}$ is the total energy of NO in the gas phase. Pd* refers to either $Pd^+$, $Pd^+H^+$, or $Pd^{+2}$.

**Pd-exchanged CHA**. We start by reporting our results for the location of Pd cations exchanged into CHA. CHA has a single T-site, which limits the number of unique Al pairs. As mentioned previously, Eqs. 1, 2 are used to evaluate the stability of the sites. This requires, for each unique Al arrangement, finding the minimum energy of the compensating protons and the Pd cation. An example of the search results for the optimum proton location (based on sampling different initial positions for a given Al siting), is shown in Supplementary Fig. 7. Some locations can be more favorable than others by as much as 1 eV. Based on an examination of the stability of the 12 different Al arrangements in CHA, we do not find a clear indication of why certain positions stabilize protons more than others. One descriptor we find useful is the distance between the oxygen atoms where the protons adsorb ($d_{O-O}$) (Supplementary Figs. 8–12). If the oxygen atoms are too close or too far (relative to Al–Al distance), the associated 2 proton configuration is not favorable. Intermediate distance almost always yields the most favorable arrangement. This descriptor can reduce the number of required calculations; however, it is not a substitute for performing the search through an approach such as the high-throughput screening employed here, especially since many candidates have similar values of $d_{O-O}$.

In addition, sample search results for the global minimum of $Pd^+H^+$ exchange are shown in Supplementary Fig. 13. As for the proton case discussed above, there is a large variance of the results depending on the initial position of the Pd cation and the proton. It is important to also note that the relative energy is sensitive to the proton position. For example, CHA-3-$Pd^+H^+$-17 and CHA-3-$Pd^+H^+$-21 both have the Pd at the center of the 6MR; however, the latter is 0.4 eV less stable due to the proton occupying a different location (Supplementary Fig. 14).

A summary of the energies of $Pd^+H^+$ and $Pd^{+2}$ in CHA, comparing unique Al positions, is given in Figs. 3, 4, respectively. In both cases, the stability of the Pd cation is heavily dependent on the Al positions, with energies varying by as much as several eV. The range of energies in $Pd^+H^+$ is much more closely spaced (<0.8 eV) compared to $Pd^{+2}$ (around 3 eV). Al pairs in the 6MR arrangements, especially in the NNNN position, provide the most favorable host for the Pd cation, in which case the Pd resides at the center of the 6MR (Fig. 5a, b and Supplementary Fig. 15a, b). This finding is consistent with other recent studies[28,29]. This geometry provides the most oxygen atoms in close proximity to the cation, but not too close. Surprisingly, the Pd cation is then most stable either at the isolated site or when the two Al pairs do not share the same ring. In these cases, the Pd cation mostly resides at the center of the 6MR (Supplementary Fig. 15c–f). For both $Pd^+H^+$ and $Pd^{+2}$, Al pairs in the 8MR and 4MR provide unfavorable arrangements, especially for the latter. In the 8MR (Supplementary Fig. 15g, h), unlike the 6MR, the two Al atoms

are farther apart and there is a lack of neighboring Si/O atoms to provide orbital overlap to stabilize the cation. In the case of the 4MR, the atoms are too closely spaced.

Despite some attempts to do so, we were not able to find a simple descriptor related to $\Delta E_{form}$ (e.g., Al–Al distance and Si/O atoms in close proximity) for the data contained in Figs. 3, 4. For $Pd^+H^+$, in almost all calculations, a minimum distance of 4 Å separates the Pd cation and the proton in the optimized structure

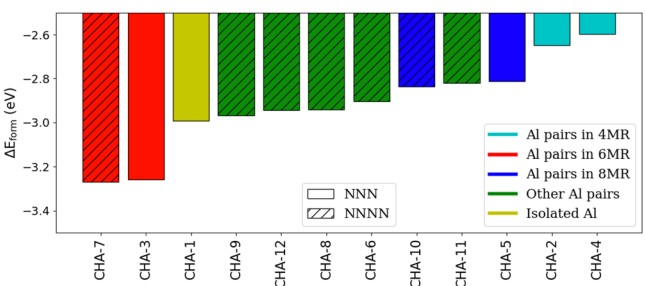

**Fig. 3 Formation Energy of Pd+H+ on CHA.** Each bar represents unique Al location(s). The color coding refers to the type of Al pairs or isolated Al in the zeolite matrix. Solid bars refer to Al pairs in an NNN configuration or the isolated site and striped bars refer to the NNNN configuration.

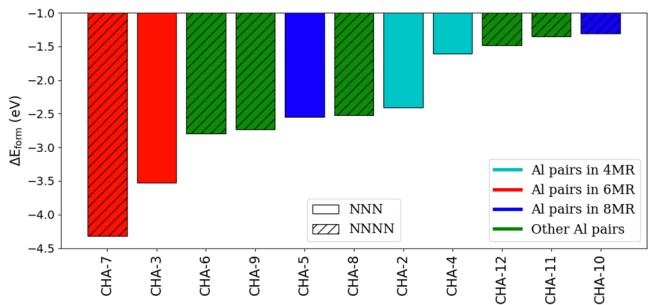

**Fig. 4 Formation energy of Pd+2 on CHA.** Each bar represents unique Al location(s). Color coding refers to the type of Al pairs or isolated Al in the zeolite matrix. Solid bars refer to Al pairs in NNN configuration and striped bars refer to NNNN configuration. The two most stable sites (CHA-7 and CHA-3) are illustrated in Fig. 5.

(Supplementary Fig. 16). This indicates repulsive interaction between the two cations at shorter distances.

Comparing $\Delta E_{form}$ between $Pd^+H^+$ and $Pd^{+2}$ on CHA shows that Al pairs are more favorable for $Pd^+H^+$ compared to $Pd^{+2}$, with the exception of Al pairs in 6MR where the order reverses. This has implications for the adsorption of guest molecules, as will be discussed later on.

**Pd-exchanged BEA.** BEA has a more diverse set of rings (4, 5, 6, and 12), 9 T sites, and is denser than CHA. This makes it more challenging to determine a full set of favorable sites for the cations. Similar to CHA, we carried out calculations for $H^+$, $H^+H^+$, $Pd^+$, $Pd^+H^+$, and $Pd^{+2}$ charge-exchanged into BEA using the approach shown in Fig. 1. Figure 6 summarizes the results for $Pd^{+2}$ in BEA, showing a rich variety of sites. It is noticeable that the four most energetically favorable structures all have Al pairs in a 6MR (images of the QM atoms are shown in Fig. 5c–f). The four structures are separated by 0.15–0.6 eV. Structure BEA-78 (Fig. 5c), the most favorable energetically, has Al pairs in a 6MR in NNN configuration. The Pd cation resides at the center of the 6MR in close proximity to 4 neighboring oxygen atoms. The three structures closest in energy (BEA-95, BEA-55 and BEA-48) have similar configurations, but differ mainly in Al placements within the 6MR and the Si/O atoms surrounding the 6MR (Supplementary Fig. 15). These four structures are followed by a number of structures where Al pairs are in a 5MR (e.g., Supplementary Fig. 17a). The most favorable 5MR structures are close in energy (separated by less than 0.15 eV). In all of these cases, the Pd cation is most stable at the center of the 5MR. In relative terms, Al pairs in a 4MR are poor hosts for the Pd cation, similar to what was observed for CHA. For Al pairs not in a 5 or 6MR, $\Delta E_{form}$ is considerably lower. The most stable structure for those cases (BEA-66, Supplementary Fig. 17b) is 1.2 eV weaker compared to the most stable structure. This reinforces the results observed in CHA (Fig. 4) where Al pairs in the same $n$-MR allow for additional stability of the cation. This likely stems from having neighboring oxygen atoms in positions favorable for orbital overlap with the Pd cation. However, this observation fails to explain why some of the other structures with Al pairs in a 6MR or 5MR arrangements have significantly smaller values of $\Delta E_{form}$. This subject will be discussed further below.

Figure 7 summarizes the formation energies for $Pd^+$ and $Pd^+H^+$ on BEA. Structure BEA-78 is the most energetically

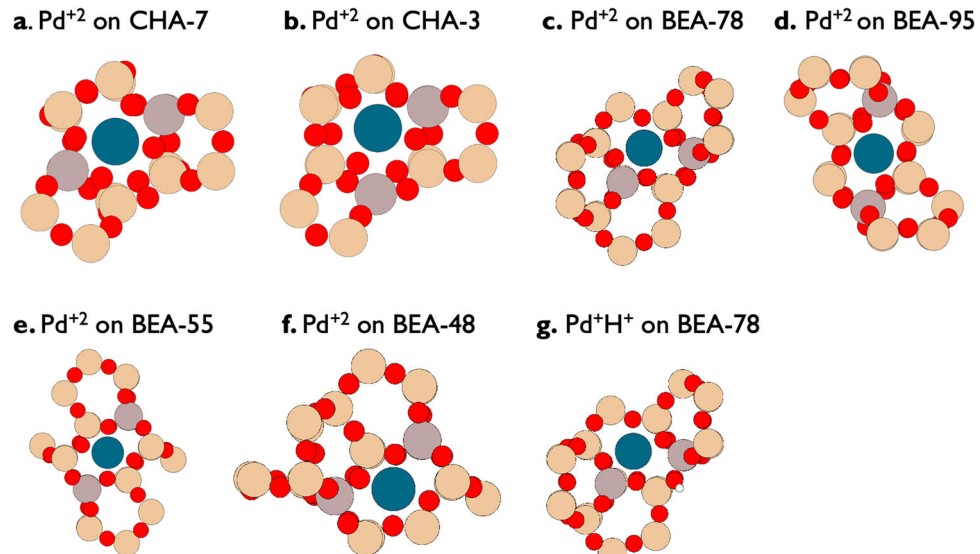

**a.** Pd+2 on CHA-7    **b.** Pd+2 on CHA-3    **c.** Pd+2 on BEA-78    **d.** Pd+2 on BEA-95

**e.** Pd+2 on BEA-55    **f.** Pd+2 on BEA-48    **g.** Pd+H+ on BEA-78

**Fig. 5 QM atoms from selected QM/MM structure optimizations.** Color coding: red = oxygen, gray = aluminum, blue = palladium, and beige = silicon.

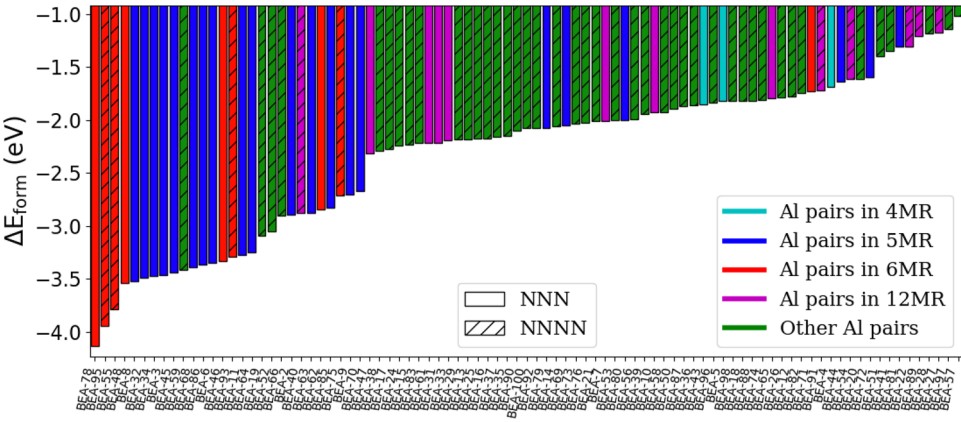

**Fig. 6 Formation energy of Pd$^{+2}$ on BEA. Each bar represents unique Al locations.** Color coding refers to the type of Al pairs in the zeolite matrix. Patterned bars refer to Al pairs in NNNN positions while solid bars refer to Al pairs in NNN positions. The four most stable sites are illustrated in Fig. 5.

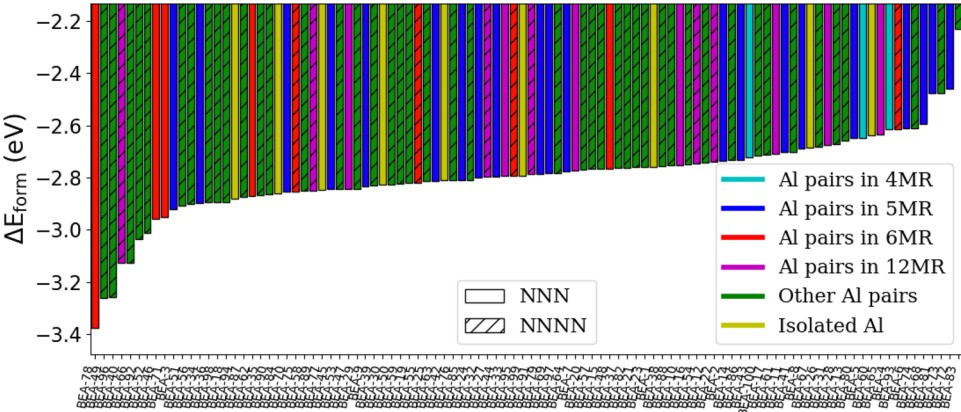

**Fig. 7 Formation energy of Pd$^{+}$/Pd$^{+}$H$^{+}$ on BEA. Each bar represents unique Al location(s).** Color coding refers to the type of Al pairs or isolated Al in the zeolite matrix. In Al pairs, patterned bars refer to Al pairs in NNNN positions and solid colors refer to Al pairs in NNN positions. The most stable site (BEA-78) is illustrated in Fig. 5.

favorable site in BEA for Pd$^{+}$H$^{+}$. It is the same site that hosts the most stable Pd$^{+2}$, for which the cation is located at the center of the 6MR and the Al pairs are in NNN arrangement (Fig. 5g). Surprisingly, structures such as BEA-49 and BEA-66 (Supplementary Fig. 17c, d), in which the Al pairs are do not share a 5MR or 6MR, are more stable than the other Al pairs in 5MR or 6MR. However, after BEA-78, and the 4 following sites, the remaining structures exhibit a nearly continuous range of energies (the difference can be <0.1 eV). Given DFT errors and the large number of structures that are very close in formation energy, it is the general picture rather than the precise order that is important. There is no clear distinction between Al pairs in the same *n*-MR observed for Pd$^{+2}$ in CHA and BEA. Some Al configurations (e.g., BEA-8) can have a high $\Delta E_{form}$ for Pd$^{+2}$ but not for Pd$^{+}$H$^{+}$. This indicates that relative stabilities are not transferable between different oxidation states. It also suggests the potential importance of sites where Al pairs do not share a 5MR or 6MR, a motif that does not generally receive much attention in computational studies, and the role they can play in adsorption/catalysis. In these structures, the presence of a neighboring Al atom in close proximity can significantly alter the $\Delta E_{form}$ compared to their respective value for isolated sites (Supplementary Fig. 18).

During the search for the most favorable charge-exchange sites, we found that in some cases the energetically most favorable Pd cation position may not involve situating the cation within a single ring, even if the Al pairs are in the same MR. Examples of

this situation are BEA-62, BEA-93, and BEA-63, shown in Supplementary Fig. 19. These positions can be >0.2 eV more stable than the Pd at the center of the 6MR. This highlights the importance of the high throughput screening approach, which can find the optimum adsorption location when it is not physically intuitive.

Given that the energetic order of Al configurations in BEA does not correlate directly between Pd +1 and +2 oxidation states, we also explored whether or not the Pd$^{+2}$ results correlate better with other divalent metal cations (Co$^{+2}$ and Ca$^{+2}$). We performed a limited number of calculations on the other cations on CHA and found some correlation with Pd$^{+2}$ results at more negative formation energies, but no consistent trend at less negative formation energies (Supplementary Fig. 20). This indicates that the results for one cation in a zeolite matrix are not readily transferable to other, even if chemically similar cations exchanged into the same zeolite. This finding is important, since the ultraviolet–visible spectrum of Co$^{2+}$ is sometimes used to identify the location of divalent cations in zeolites[42,43].

**Comparison between Pd-exchanged CHA and BEA.** In general, there are similarities in Pd-exchanged BEA and CHA. In both cases, the Pd cation (both as a Pd$^{+2}$ and Pd$^{+}$H$^{+}$) is energetically most favorable in the 6MR. For Pd$^{+}$H$^{+}$ and Pd$^{+2}$, the site with the highest $\Delta E_{form}$ has almost identical energy values

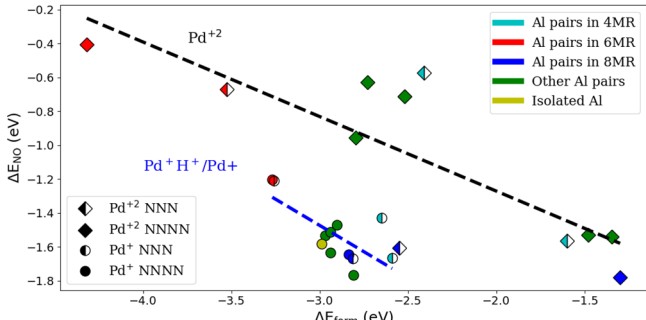

**Fig. 8 NO Binding energy versus Pd formation energy on CHA.** The color coding refers to the type of Al in the $n$-MR. The marker shapes ($\diamond$ and $\circ$) represent $Pd^{+2}$ and $Pd^+/Pd^+H^+$, respectively. Filled markers are used for Al pairs in NNNN configuration and half-filled markers are for isolated Al or Al in NNN configurations. For visual clarity, linear lines based on fitting the data are added (black and blue for $Pd^{+2}$ and $Pd^+/Pd^+H^+$, respectively).

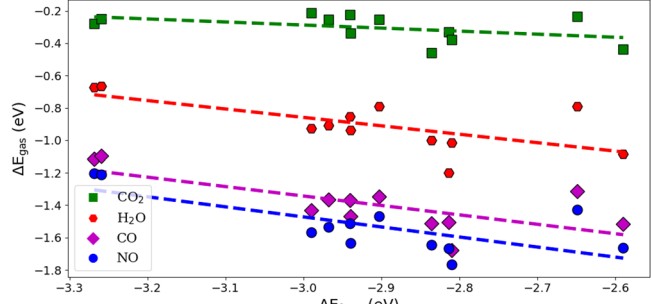

**Fig. 9 Binding energy of a number of gaseous species on Pd-exchanged CHA ($Pd^+/Pd^+H^+$) against Pd formation energy (at unique Al configurations).** Dotted lines refer to fitted data for each gas adsorbate. Detailed structures for each structure are available in the Supplementary Information.

(around $-3.3$ eV and $-4.2$, respectively). The 6MR in BEA is more oval shaped (Fig. 5c) compared to that of CHA (Fig. 5a), placing the oxygen atoms at a closer distance. The four oxygens close to the Pd cation in the 6MR are on average located 2.0 Å from the Pd in BEA and 2.2 Å in CHA. This, however, does not seem to impact the formation energy.

**NO adsorption.** The location of Pd cations in CHA and BEA has practical implications for the capability of the zeolite to act as a passive NO$_x$ adsorber (PNA) during the cold start of automobiles. The nature of the oxidation state of the Pd cations and their location in the zeolite continue to be debated in the scientific literature[28–30]. Here, we attempt to shed some light on the subject based on the results of our high-throughput screening. We focus on NO adsorption on Pd-exchanged-CHA, considering $Pd^+$, $Pd^+H^+$, and $Pd^{+2}$. Figure 8 shows the correlation between calculated values of $\Delta E_{NO}$ and $\Delta E_{form}$. Generally, a weaker $\Delta E_{form}$ correlates with a stronger NO adsorption energy. Similar to most adsorption processes, the stronger the binding energy of the cation site, the less electron density is available to bind the guest gas molecule[44]. There appears to be a linear correlation with a low mean absolute error (MAE) for NO adsorption on $Pd^+H^+$-exchanged CHA (0.1 eV); however, this is not the case for $Pd^{+2}$-exchanged CHA.

Figure 8 demonstrates that there is a clear distinction in NO adsorption between $Pd^+H^+$ and $Pd^{+2}$. For a similar $\Delta E_{form}$, $\Delta E_{NO}$ in $Pd^+H^+$ is much stronger (by as much as 1 eV). For most of the Al pair arrangements, the NO binds more strongly to $Pd^+H^+$ compared to $Pd^{+2}$. We also find NO adsorption on $Pd^+H^+$ to be stronger when the $Pd^+$ and NO unpaired electrons are paired (compared to two unpaired electrons) (Supplementary Table 2). This might be one reason for the stronger binding on $Pd^+H^+$ (given that NO adsorption on $Pd^{+2}$ has an unpaired NO electron).

Previous literature work showed that several factors affect NO adsorption on Pd-exchanged zeolite (e.g., how Pd is introduced to the zeolite support)[45–47]. Isolated Pd cations have been shown to be desired for PNAs[46,48–50]. Experimentally, there are direct evidence of $Pd^+$ presence in the zeolites, as shown in the electron paramagnetic resonance spectroscopy (EPR) results[48,51,52]. Although it is unclear if $Pd^+H^+$ is present under operating conditions; however, Fig. 8 indicates that if present, it is a superior NO adsorption site compared to $Pd^{+2}$.

Figure 8 also shows that Al pairs in the 6MR arrangement (especially in NNNN configuration), which has been discussed most extensively in the literature because they hold $Pd^{+2}$ cations

most stably, are weaker sites for NO adsorption compared to other Al arrangements. This is not unexpected since the more stable the Pd, the more weakly it can bind to a guest molecule. It is striking, however, how large the difference in $\Delta E_{NO}$ is compared to the other Al arrangements (0.25–1.5 eV). Figure 8 also highlights how many of the CHA sites are very close in energy, especially for $Pd^+H^+$. Therefore, it is very difficult based on the small differences in energy values to discern spectroscopic data and assign them to specific sites. The results indicate that an ensemble of sites of very similar energies can contribute similarly to the adsorption of NO.

We further extended our work to analyze the selectivity of Pd-exchanged-CHA toward NO compared to other gaseous species present in automobile exhaust (namely CO, $CO_2$, and $H_2O$). We limit the calculations to $Pd^+$ and $Pd^+H^+$ exchanged CHA based on its much stronger NO binding energy (Fig. 8). The results presented in Fig. 9 show that other adsorbates follow a linear scaling relation, similar to that for adsorbed NO (MAE for each specie is <0.1 eV). Among the studied gaseous species, NO adsorbs the most strongly to Pd-exchanged-CHA. CO also adsorbs strongly, only 0.1 eV less than NO. Water and $CO_2$ adsorb much more weakly (on average 0.6 and 1.2 eV less than NO, respectively), showing that Pd-exchanged-CHA is an excellent adsorbent at selectively removing NO from the other non-toxic gases in automobile exhaust. We also observe that as the Pd formation energy decreases, the gap in adsorption energy between NO and the other species becomes more pronounced, likely a result of the Pd electron density being mostly used in binding to the zeolite framework.

Finally, we note that the DFT-based high-throughput screening reported here provides a robust systematic path for identifying the most energetically favored cation exchange sites. While the method is applied here to zeolites, it could be extended to identify the energetically preferred location of cations in many other materials (e.g., post-synthetic modifications of MOFs).

## Methods

**Theoretical calculations.** A hybrid quantum mechanics/molecular mechanics (QM/MM) approach was used to model the zeolite structure. Detailed implementation of this model can be found elsewhere[17]. The QM/MM approach has proven to account for long-range Coulombic and dispersive interactions, which are critical in describing the zeolite framework interactions with adsorbates[40] and has been widely used to model zeolite catalysis/adsorption[32,37–39,53–63]. Many studies have shown that the QM/MM approach gives a good prediction of experimental data for different zeolites and adsorbates[17,32,37–40,57–59,61–63]. This approach is also computationally more efficient than periodic calculations since it requires a smaller number of QM atoms, especially when the unit cell contains a large number of atoms (e.g., BEA and FAU). A number of studies benchmarked QM/MM calculations against periodic calculations and found similar results[31,53]. All QM/MM calculations were done with a development version of Q-Chem[64,65].

The adsorbate(s) and a cluster encompassing the active site are described by QM and the rest of the zeolite is modeled by MM using a standard force field of the CHARMM type with the P2 parameter set[18]. During structural optimization, the QM region is allowed to relax while the MM region is fixed. The B97-D3 exchange functional[66], a GGA exchange functional, is used as an initial filter to determine the optimum site of the cation followed by calculations done with the more accurate range-separated hybrid functional, $\omega$B97X-D[67], which was shown to be among the best performing hybrid functionals in a benchmarking study[19]. The appropriate def2 effective core potential was used on the Pd atom. For each structure, the def2-SV(P) basis set was used to obtain the optimized structure geometry, and further energy refinement was done using a single-point calculation at the def2-TZVPD level of theory[68].

**Zeolite model.** The crystallographic structure of CHA and BEA were obtained from the International Zeolite Association (IZA) database[69]. Cluster models containing 696 and 810 tetrahedral atoms (T696 and T810) were used to model CHA and BEA topologies, respectively (Supplementary Fig. 21). The CHA cluster is based on the previous work from or group[28]. While earlier work has suggested that a 100 T-atom cluster model is sufficient[37], larger cluster models are used here due to the marginal additional computational cost and the extended active site region in some of the calculations. Each cluster was terminated with hydrogen atoms replacing terminal oxygen atoms.

## Data availability
The raw data that support the results within this paper and other findings of this study are available at the zenodo repository (https://doi.org/10.5281/zenodo.6509562).

## Code availability
The Q-Chem code (version 5.3; www.q-chem.com) was used in standard form for all electronic structure calculations. External custom-written python code was used to generate Q-Chem jobs for the high-throughput workflow; this code is available via the zenodo repository (https://doi.org/10.5281/zenodo.5978699).

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

## Acknowledgements

H.A. is greatly thankful to Saudi Aramco for their funding. This material is based upon work supported by the U.S. Department of Energy's Office of Energy Efficiency and Renewable Energy (EERE) under the Vehicle Technologies Program Award Number DE EE0008213. M.H.-G. acknowledges support from the Director, Office of Science, Office of Basic Energy Sciences, of the U.S. Department of Energy through the Gas Phase Chemical Physics Program, under Contract No. DE-AC02-05CH11231. This research used resources of the National Energy Research Scientific Computing Center, a DOE Office of Science User Facility supported by the Office of Science of the U.S. Department of Energy under Contract No. DE-AC02-05CH11231. The authors also thank UC Berkeley's Molecular Graphics and Computation Facility (supported by NIH S10OD023532) for their computational resources. This work used the Extreme Science and Engineering Discovery Environment (XSEDE) Comet at the service-provider through allocation TG-CE200085. We also thank Dr. Jeroen Van der Mynsbrugge for many productive discussions.

## Author contributions

H.A. and A.T.B. conceptualized the project. H.A., M.H.G., and A.T.B. developed the methodology. H.A. wrote the code, performed the calculations, and wrote the original draft. All authors participated in data analysis and editing the paper. M.H.G. and A.T.B. supervised the research reported in the paper.

## Competing interests

M.H.G. is a part-owner of Q-Chem Inc, whose software was used for the calculations reported here. H.A. and A.T.B. declare no competing interests.
