## [Peer Review File · Nature Communications]

Reviewers' comments:

Reviewer #1 (Remarks to the Author):

The paper presents an approach towards identifying the energetically most favorable adsorption site for a cation using density functional theory. It addresses an important gap to identify the most favorable site among other sites via a systematic methodology to enumerate all possibilities since past literature has focused on experimentally observed Al-substitution sites or those observed with experience. The method proposed identifies possible substitution sites for Aluminum by enumerating single substitution sites and then expanding the enumeration towards identifying possible further substitutions. Further, for each of the enumerated possibilities of substitution, a quantum mechanics/molecular mechanics approach to identify the feasible site for the cation location is used and the energy calculations are performed to identify the most favorable site.

There are several questions that need to be addressed before the work is publishable. Overall, the claim of the authors to consider all Al-substituted sites is not true since they reduce the enumerations only to the 4th nearest neighbor with no strong argument to drop further Al substitutions while literature considers further Al substitutions as well [3]. Also, exceptions for the Loewenstein's rule exist and they have not been considered in this approach [1-2]. The methodology needs to be explained in more detail for the work to be reproducible and almost all figures need a revision for better understanding. Several of the plots are used for inferences which have major deviations and thus cannot be used to arrive at conclusions. Following are the comments that need to be worked on including major and minor comments.

1. The paper claims the methodology to be high throughput but does not provide enough evidence to support this claim. The study is focused in detail on the substitution of Pd in CHA and BEA, however, it is performed only on these 2 zeolites and for the method to be claimed as high throughput, following can be done:
 - a. Showcase the applicability of the proposed method on several zeolites thus justifying the high throughput claim on the scale of the study performed.
 - b. Perform the study on a zeolite where the number of possible Al-substitutions are significantly higher (CHA and BEA have $O(10^2)$ enumerated substitutions) justifying the claim for the method to be high-throughput.
2. The methodology to generate all possible Al pairs needs to be explained in more detail. Also explain figure S3 since several redundancies are possible following this algorithm. Especially, how have the authors ensured the 3rd checklist i.e., 'no sub-list belongs in a smaller MR'. Figure S3 needs to be explained in much more detail to ensure reproducibility.

3. Authors have ignored all Al-pairs that are further apart than NNNN. This assumption leads to reduction in the computational complexity. Please look at ref [3] for cases where Al-pairs are much more distant.
4. How is the molecular graph generated (Page 6, 2nd paragraph)? Citations 30, 31 are software to build zeolite lattices and more details need to be given as to how the molecular graph is generated.
5. The authors need to add the methodology/data for nuclear repulsion energy calculations in SI that enabled the reduction of unique structures for CHA(26) and BEA(212). The total numbers before this reduction also need to be given.
6. Structures with Al pairs not in the same MR but on opposite sides of the open cage were not considered. Why is this so? If the process is high throughput, it should be accounting for all of the sites or a strong justification needs to be given for ignoring certain sites. A study showing that these structures were evaluated and discovered to be not relevant for investigation should clarify things. The authors have done a good job explaining the assumption on page 8 at the citations [11, 32-35] and a similar study in this case should clarify things.
7. Figure S6: 32 structures' results should be present. Report findings for all 32.
8. Figure S7: at same O-O distance, wide range of eV is observed. Annotate the figure and add which structures were leading to which points. Not necessary for all points but for maybe O-O between 5-6, showcase the figures so an understanding can be developed as to what difference in the structure led to the range of values.
9. Figure S8: For 2 Al sites, 32 possible structures should exist. Only 26 have been shown. Plot results for all.
10. Figure 6: Al pairs from the 12 MR are not observed in any results. Since Al pairs with greater connectivity than NNNN were dropped from the study, some of the possible pairs of NNNNNN (diametrical opposites) of the 12 MR are not considered. In a high throughput study, these cases should not be ignored, since the goal of computational studies is to explore all options albeit a strong argument exists not to (e.g., Loewenstein's rule for not considering N. Even though exceptions exist, this can be considered).
11. Figure 9: Data shows that the outliers are significant, and comparisons or trends may be studied but conclusions on each category should not be drawn based on this plot. Legend needs to be clear as well.
12. Conclusion: Not all Al pair arrangements are considered.
13. Figure S11: Legend for squares and circles.
14. Figure S13: R-squared values need to be reported for parity plots.
15. Other experimental work regarding the stability of Al substitutions in CHA and BEA can be used for validation [4,5].

16. Cite relevant work on QM/MM model from all sources and not just past work from the research group: [6, 7, 8]

17. Other minor comments:

a. Page 4 – Add GGA abbreviation.

b. The data for all generated structures and optimized geometries are missing from the SI.

c. Legends of graphs need to be inclusive and there should be no need to read captions to read legends of a plot. (solid color vs crossed color, shape of points, etc.).

d. Abbreviation of UV-V on page 18.

e. Grammar – end captions with periods.

f. Contrasting colors might make plots more legible. Gray color might be changed for a different color like cyan, orange, etc.

g. Acknowledgement: ...like to thank Saudi Aramco...

References:

1. Fletcher, R. E.; Ling, S.; Slater, B. Violations of Lowenstein's rule in zeolites, *Chem. Sci.* 2017, 8, 7483-7491.

2. Tang, X., Liu, Z., Huang, L., Chen, W., Li, C., Wang, G., ... & Zheng, A. (2019). Violation or Abidance of Löwenstein's Rule in Zeolites Under Synthesis Conditions?. *ACS Catalysis*, 9(12), 10618-10625.

3. Di Iorio, J. R., Li, S., Jones, C. B., Nimlos, C. T., Wang, Y., Kunkes, E., ... & Gounder, R. (2020). Cooperative and Competitive Occlusion of Organic and Inorganic Structure-Directing Agents within Chabazite Zeolites Influences Their Aluminum Arrangement. *Journal of the American Chemical Society*, 142(10), 4807-4819.

4. Bortnovsky, O., Sobalik, Z., & Wichterlová, B. (2001). Exchange of Co (II) ions in H-BEA zeolites: identification of aluminum pairs in the zeolite framework. *Microporous and mesoporous materials*, 46(2-3), 265-275.

5. Nishitoba, T., Yoshida, N., Kondo, J. N., & Yokoi, T. (2018). Control of Al distribution in the CHA-type aluminosilicate zeolites and its impact on the hydrothermal stability and catalytic properties. *Industrial & Engineering Chemistry Research*, 57(11), 3914-3922.

6. Nieminen, V., Sierka, M., Murzin, D. Y., & Sauer, J. (2005). Stabilities of C3–C5 alkoxide species inside H-FER zeolite: a hybrid QM/MM study. *Journal of Catalysis*, 231(2), 393-404.

7. Lyne, P. D., Hodoscek, M., & Karplus, M. (1999). A hybrid QM– MM potential employing Hartree–Fock or density functional methods in the quantum region. *The Journal of Physical Chemistry A*, 103(18), 3462-3471.

8. Hillier, I. H. (1999). Chemical reactivity studied by hybrid QM/MM methods. *Journal of Molecular Structure: THEOCHEM*, 463(1-2), 45-52.

Reviewer #2 (Remarks to the Author):

Recommendation: Rejection

Comments:

The authors present a workflow to identify favorable cation-exchanged sites in zeolites. They systematically investigate 696-T CHA and 810-T BEA cluster models with different arrangements of Al pairs. They use QM/MM calculations to identify favorable Pd-exchange sites, and their corresponding influence on NO adsorption performance. However, the novelty of this manuscript is very limited because the whole workflow is simply a combination of conventional approaches. The authors claim this methodology is easily applicable and transferable to any adsorption/catalysis problem with any zeolite. This is not true. This workflow requires comprehensive computational resources, and it cannot enumerate all possible ways of Al incorporation and Pd-exchanging periodic models, which are closer to real zeolites than cluster models. To reduce the computational costs, the authors significantly decrease the Al contents in these models, which are far away from real zeolites. Therefore, this approach is only applicable to zeolite structures with unrealistically low Al contents. I would recommend rejection as the novelty of this manuscript does not meet the high standard of *Nat. Commun.* Detailed comments include:

1. The reported Si/Al ratio in Pd-exchanged SSZ-13 was 8.9 (ref. 28), which corresponds to 3.6 Al atoms per CHA unit cell (36T). This manuscript only considers 2 Al atoms in a 696-T CHA cluster model, neglecting a large number of possible Al arrangements. It has been reported in literature that the variation in the content of heteroatoms in zeolite frameworks significantly influence the distribution of extra-framework cations. So, 2 Al atoms in a 696-T cluster cannot represent the real situation, where 3.6 Al atoms are distributed in a 36-T unit cell.
2. The poor data in Figure S7 is not sufficient to support the authors' claim that intermediate O-O distance yields more favorable arrangements.
3. Please provide structure illustrations of all the structures mentioned in this manuscript, such as BEA-36, BEA-59, and BEA-62.

4. Molecular dynamic method is recommended for NO adsorption calculations.
5. Some unfavorable charge-exchanged sites (e.g. Pd⁺ or H⁺ at 4 or 5 MR) that are inaccessible in the NO adsorption process can be excluded before QM/MM calculation to reduce computational cost.

Reviewer #3 (Remarks to the Author):

The manuscript "Assessing the stability of Pd-exchanged sites in zeolites with the aid of a high throughput quantum chemistry workflow", by Hassan Aljama, Martin Head-Gordon, and Alexis T. Bell, describes a DFT study performed to elucidate the distribution of Pd⁺ and Pd²⁺ cations in two Al-containing zeolites, CHA and BEA.

Although the manuscript is well written, the results are relevant, and the discussion and conclusions are interesting, I do not think the paper deserves publication in Nature Communications, since the results do not have the high standard of novelty and relevance required for this journal. I would recommend submitting it to a more specialized journal, such as Journal of Chemical Theory and Computation.

But before submitting I would like to point some important points that, in my opinion, should be taken into account:

- 1) There have been several computational studies on the distribution of cations in Al-containing zeolites, such as those of carried out by Sauer, Dedecek, Sklenak, Nachtigall, Newsam, Mellot, Cheetham, Ruiz-Salvador, Sastre, Catlow, etc. These studies have provided a large body of knowledge in the field, which has not been discussed neither in the introduction nor in the discussion. The present study gives a feeling of isolation, since the results are not put into the context of the state of the art; the authors should present an analysis of the relation between the present results and relevant results present in the literature, mentioning how their results agree or disagree with previous results.

- 2) The different configurations for the distribution of cations are obtained using an energy-based method, but a more rigorous approach would be to use cation distribution methods based on the symmetry of the crystal, such as those implemented in the programs SOD or Supercell. These

methods could help not only to study the distribution of Al atoms, but also in the distribution of extra-framework cations.

3) The authors use QM/MM models to study the zeolites. This approach has been used due to its computational cost. But recent advances both in hardware and software make feasible the study of a large number of configurations using purely QM periodic calculations. For instance, I guess that the present study could have been carried out using CP2K, which would have allowed the proper inclusion of periodicity and long-range interactions, and the errors introduced by spurious interactions between the QM and MM regions could have been avoided. It is now too late to do the calculations, but the authors should mention how much they estimate that these errors could be.

4) This comment is somehow related to the previous one. Many experimental and theoretical studies have shown that the introduction of extra-framework cations does have a relevant effect in terms of distorting the zeolite structure (in some cases even leading to the collapse of the structure), which in turn might change the positions of the cations themselves. This effect is largely neglected in the present study, since the atoms in the MM region are fixed, so the authors should at least mention how the results are likely to be affected by this neglect.

5) Finally, the manuscript has several typos and minor language errors that should be corrected.

Response to reviewers' comments

Reviewer #1:

The paper presents an approach towards identifying the energetically most favorable adsorption site for a cation using density functional theory. It addresses an important gap to identify the most favorable site among other sites via a systematic methodology to enumerate all possibilities since past literature has focused on experimentally observed Al-substitution sites or those observed with experience. The method proposed identifies possible substitution sites for Aluminum by enumerating single substitution sites and then expanding the enumeration towards identifying possible further substitutions. Further, for each of the enumerated possibilities of substitution, a quantum mechanics/molecular mechanics approach to identify the feasible site for the cation location is used and the energy calculations are performed to identify the most favorable site.

Response: We thank the referee for the encouraging summary. In the manuscript, we have clarified that the objective is to determine the favorable cation/proton sites for each competitive Al configuration, rather than the Al distribution itself. Most Al configurations are not too difficult to model theoretically; however, the associated cation site (for a given Al configuration) is what is difficult to discern experimentally and challenging to predict theoretically. We made the following change to the manuscript on page 3 to clarify this point:

“With this trend in mind, the objective of this work is to present and apply a DFT-based computational framework to identify the energetically most favorable adsorption sites (cation or proton) in any zeolite using a systematic high-throughput approach. The underlying calculations employ a hybrid quantum mechanics/molecular mechanics (QM/MM) model [17,18] to capture extended environment around each active site, using high quality density functionals [19] to attain acceptable accuracy. Our approach starts by evaluating the possible zeolite structures that arise for different Si/Al ratios, then focuses on searching for energetically favorable cation/proton(s) adsorption sites for each structure using a lower level of theory (e.g. smaller basis set and cheaper functional) as a filter. The most favorable cation adsorption sites are then evaluated at a higher level of theory.”

There are several questions that need to be addressed before the work is publishable. Overall, the claim of the authors to consider all Al-substituted sites is not true since they reduce the enumerations only to the 4th nearest neighbor with no strong argument to drop further Al substitutions while literature considers further Al substitutions as well [3]. Also, exceptions for the Loewenstein's rule exist and they have not been considered in this approach [1-2].

Response: We do not claim that our work considers all Al-substituted sites since we limit the Al pairs to only the NNNN neighbors, as the reviewer points out. We revisited the wording in the manuscript to ensure there is no claim of this sort. While we consider all Al pairs in a 3N and 4N configuration (the examples in the paper are given for CHA and BEA), the approach we use, as explained in Figure S3, is valid for any zeolite material. We also revisited the assumption related to the 5N configuration. We have tested this assumption by doing several calculations on Al pairs in 5N configurations. We found that in all 5 cases, the Pd formation energy is weaker compared to

the other 3N and 4N configurations. This further supports our point that Al pairs further than 4N configurations have a minimal effect on each other and begin to behave as isolated Al atoms. To clarify this point, we have added the following text to page 6 of the revised manuscript.

Pairs involving Al atoms farther apart than an NNNN configuration were not considered since as the Al-Al distance increases, the impact of having the Al pair diminishes and each atom starts behaving as an isolated Al site (some tests demonstrating this point are shown in Figures S1 and S2).

Figure S1: Images of the optimized geometry of 5 different Pd²⁺-exchanged-CHA structures where the Al pairs are in a 5N configuration. The same color coding as in Figure 5, however, for clarity purposes, Al atoms are enlarged and are shown in green. The calculated formation energies of the structures are a. -0.72 b. -0.72 c. -0.84 d. -1.05 and e. -1.22 eV.

Figure S2: Pd²⁺-exchanged-CHA formation energy as a function of Al-Al distance. Color coding refers to the type of Al pairs.

Regarding Al pairs that are direct neighbors, we acknowledge the existence of few cases where the Loewenstein's rule is violated. However, these cases remain the outliers and for most cases, these sites are not present. We added the following text on page 6 of the revised manuscript:

“Exceptions to Loewenstein’s rule have been reported in the literature; however, these cases remain an exception and not the norm [34,35]”

The methodology needs to be explained in more detail for the work to be reproducible and almost all figures need a revision for better understanding. Several of the plots are used for inferences which have major deviations and thus cannot be used to arrive at conclusions. Following are the comments that need to be worked on including major and minor comments.

We thank the reviewer for this guidance, and address the specific suggestions made by the reviewer in the order that they appear below.

1. The paper claims the methodology to be high throughput but does not provide enough evidence to support this claim. The study is focused in detail on the substitution of Pd in CHA and BEA, however, it is performed only on these 2 zeolites and for the method to be claimed as high throughput, following can be done:

- a. Showcase the applicability of the proposed method on several zeolites thus justifying the high throughput claim on the scale of the study performed.
- b. Perform the study on a zeolite where the number of possible Al-substitutions are significantly higher (CHA and BEA have $O(10^2)$ enumerated substitutions) justifying the claim for the method to be high-throughput.

Response: High throughput refers to being able to perform a significantly greater number of calculations in a systematic way compared to conventional methods. In typical DFT studies of zeolites, due to the many possible permutations, simplifications are made to consider a limited number of active sites based on physical intuition and/or available experimental data. To make this point, we have added the following text to page 3 of the revised manuscript.

“For these reasons, many higher-level theoretical studies of zeolite-catalyzed reactions [9-12] using density functional theory (DFT) have chosen to focus on a few active sites selected on the basis of limited experimental evidence and/or physical intuition”

Our work includes over 7,000 explicit DFT calculations. In most studies (e.g. refs. [13, 14, 16, 30-32]) the number of calculations does not exceed a maximum of 500. This > 10-fold increase in the number of explicit calculations, together with the systematic selection algorithm, qualifies our work as high throughput, in our view. To address this point we have added the following text to the revised manuscript on page 4:

“This approach allowed us to perform over 7,000 explicit DFT calculations in a systematic way, which is more than an order of magnitude higher than what is done typically in DFT studies of zeolites [13, 14, 16, 30-32].”

CHA is one of the simplest zeolites since it has only one unique T-site. However, BEA remains one of the most challenging zeolites to study theoretically due to the presence of 9 unique T-sites. To the best of our knowledge, no theoretical study has thoroughly investigated BEA due to its complexity. We have thoroughly studied Pd⁺, Pd²⁺, Pd⁺H⁺, H⁺, and H⁺H⁺ cation-exchanged in BEA, showing the versatility of our approach. There are a number of studies that focus on similarly complex zeolites

(e.g. MFI); however, as we have indicated in the manuscript, many simplifications are made to focus on a small number of active sites and no systematic attempt was made to thoroughly search for the most favorable site, as we have done in our work (see the two references below).

- Phys. Chem. Chem. Phys., 2019, 21, 18758-18768 [ref 16 in the manuscript]
- Catalysis Communication 2014, 52, 98-102 [ref 13 in the manuscript]

2. The methodology to generate all possible Al pairs needs to be explained in more detail. Also explain figure S3 since several redundancies are possible following this algorithm. Especially, how have the authors ensured the 3rd checklist i.e., ‘no sub-list belongs in a smaller MR’. Figure S3 needs to be explained in much more detail to ensure reproducibility.

Response: We agree that this aspect of our manuscript should be improved. However, to clarify our work, we have therefore added the following text on page S6 of the revised manuscript:

“Sub-lists are created in the same manner as a list. They start with the same initial Al atom, and include elements of length 4 to X. If a sub-list satisfies the first two checks (first and last elements are the same, and no element appears twice except the first and last ones), then a sub-list forms an X MR. Thus, the considered list does not form a MR.”

3. Authors have ignored all Al-pairs that are further apart than NNNN. This assumption leads to reduction in the computational complexity. Please look at ref [3] for cases where Al-pairs are much more distant.

Response: Indeed, reference [3] (given by the reviewer) shows the existence of Al pairs further apart than NNNN. In our previous submission, we assumed that for a very large separation between the Al atoms comprising a pair, each Al would behave as an isolated Al. We had assumed that this begins taking place at 4N (where the two Al atoms are separated by two Si atoms). Per the recommendation of the reviewer, we have revisited and tested this assumption by doing the calculations on 5 different CHA structures where Al pairs are in a 5N configurations. In each unique Al pair, a search for the most stable cation position was done. We found that in all 5 cases, Pd⁺² formation energy of each site is weaker than any of the studied Al pairs in this work (both 3N and 4N configuration). The revised paper therefore includes the following statement on page 6:

“Pairs involving Al atoms farther apart than an NNNN configuration were not considered since as the Al-Al distance increases, the impact of having the Al pair diminishes and each atom starts behaving as an isolated Al (some tests demonstrating this point are shown in Figures S1 and S2)”

Figure S1: Images of the optimized geometry of 5 different Pd²⁺-exchanged-CHA structures where the Al atoms comprising the pair are in a 5N configuration (separated by 3 Si atoms). The color coding is the same as in Figure 5, however, for clarity, the Al atoms are enlarged and are shown in green. The calculated formation energies of the structures are a. -0.72 b. -0.72 c. -0.84 d. -1.05 and e. -1.22 eV.

For CHA, we have added a plot of the Pd formation energy as a function of Al-Al distance based on these new calculations. The results (shown in Figure S2), demonstrate that the greater the Al pair distance, the more unfavorable the energy for forming that cation-exchanged site.

Figure S2: Pd²⁺-exchanged-CHA formation energy as a function of Al-Al distance. Color coding refers to the type of Al pairs

4. How is the molecular graph generated (Page 6, 2nd paragraph)? Citations 30, 31 are software to build zeolite lattices and more details need to be given as to how the molecular graph is generated.

Response: A molecular graph was generated based on the library created by a group at Ghent University. We have modified the citation to “T. Verstraelen, Molmod Software Library, <http://molmod.ugent.be/software>”, as recommended by the original creators of molmod.

5. The authors need to add the methodology/data for nuclear repulsion energy calculations in SI that enabled the reduction of unique structures for CHA(26) and BEA(212). The total numbers before this reduction also need to be given.

Response: We have included full information (structures and energies) on the nuclear repulsion energy in the SI. We also made the following modification to the manuscript (page 8):

“Since the procedure described above can produce duplicate structures, we relied on calculating the nuclear repulsion energy (full results are available in the SI).”

6. Structures with Al pairs not in the same MR but on opposite sides of the open cage were not considered. Why is this so? If the process is high throughput, it should be accounting for all of the sites or a strong justification needs to be given for ignoring certain sites. A study showing that these structures were evaluated and discovered to be not relevant for investigation should clarify things. The authors have done a good job explaining the assumption on page 8 at the citations [11, 32-35] and a similar study in this case should clarify things.

Response: In our previous submission, we did not consider Al pairs where the atoms are on opposites side of the 12MR in BEA. In preparing this revised version, we have revisited this assumption by doing the calculations on all those structures (14 in total) and have included their data in Figures 6 and 7. We have also added color coding to Al pairs in the 12 MR to distinguish them from the other Al pairs. The most stable sites, however, remain the same as was presented earlier. Please note that the numbering of structures has changed (all structures are available in the SI).

Figure 6: Formation energy of Pd²⁺ on BEA. Each bar represents unique Al locations. Color coding refers to the type of Al pairs in the zeolite matrix. Patterned bars refer to Al pairs in NNNN positions while solid bars refer to Al pairs in NNN positions. The four most stable sites are illustrated in Figure 5.

Figure 7: Formation energy of Pd⁺/Pd⁺H⁺ on BEA. Each bar represents unique Al location(s). Color coding refers to the type of Al pairs or isolated Al in the zeolite matrix. In Al pairs, patterned bars refer to Al pairs in NNNN positions and solid colors refer to Al pairs in NNN positions. The most stable site (BEA-78) is illustrated in Figure 5.

7. Figure S6: 32 structures’ results should be present. Report findings for all 32.

Response: Our manuscript may not have been clear previously, but for optimal proton location, the maximum number of calculations is only 16 (compared to Pd⁺H⁺ where the number is 32). This is now clearly stated in the revised manuscript on page 9:

“For structures with an Al pair, finding the optimum position of a Pd²⁺ requires 8 calculations per structure, and finding the optimum position for H⁺H⁺ requires 16 total calculations.”

8. Figure S7: at same O-O distance, wide range of eV is observed. Annotate the figure and add which structures were leading to which points. Not necessary for all points but for maybe O-O between 5-6, showcase the figures so an understanding can be developed as to what difference in the structure led to the range of values.

Response: We have revised the plot and added 4 other plots for other zeolite structures. In each plot, we have inserted images representative structures for short, intermediate, and long O-O distances.

Figure S8: CHA-2 optimum proton locations search results. Each data represents a different initial position of the protons. The x-axis refers to the O-O distance (where the oxygen is the atom H adsorbs on) in the optimized structure and the y-axis is the energy of the optimized structure relative to the most stable structure (CHA-2-H⁺H⁺-14). Insert images of select structures are shown (a. CHA-2- H⁺H⁺-2 b. CHA-2-H⁺H⁺-14 c. CHA-2- H⁺H⁺-7). Structures of all the other data points are available in the ESI.

Figure S9: CHA-3 optimum proton locations search results. Each data represents a different initial position of the protons. The x-axis refers to the O-O distance (where the oxygen is the atom H adsorbs on) in the optimized structure and the y-axis is the energy of the optimized structure relative to the most stable structure (CHA-3- H⁺H⁺-3). Insert images of select structures are shown (a. CHA-3- H⁺H⁺-1 b. CHA-3- H⁺H⁺-3 c. CHA-3- H⁺H⁺-11). Structures of all the other data points are available in the ESI.

Figure S10: CHA-5 optimum proton locations search results. Each data represents a different initial position of the protons. The x-axis refers to the O-O distance (where the oxygen is the atom H adsorbs on) in the optimized structure and the y-axis is the energy of the optimized structure relative to the most stable structure (CHA-5- H⁺H⁺-3). Insert images of select structures are shown (a. CHA-5- H⁺H⁺-7 b. CHA-5- H⁺H⁺-3 c. CHA-5- H⁺H⁺-4). Structures of all the other data points are available in the ESI.

Figure S11: CHA-6 optimum proton locations search results. Each data represents a different initial position of the protons. The x-axis refers to the O-O distance (where the oxygen is the atom H adsorbs on) in the optimized structure and the y-axis is the energy of the optimized structure relative to the most stable structure (CHA-6- H⁺H⁺-4). Insert images of select structures are shown (a. CHA-6- H⁺H⁺-2 b. CHA-6- H⁺H⁺-4 c. CHA-6- H⁺H⁺-8). Structures of all the other data points are available in the ESI.

Figure S12: CHA-8 optimum proton locations search results. Each data represents a different initial position of the proton. The x-axis refers to the O-O distance (where the oxygen is the atom H adsorbs on) in the optimized structure and the y-axis is the energy of the optimized structure relative to the most stable structure (CHA-8- H⁺H⁺-1). Insert images of select structures are shown (a. CHA-8- H⁺H⁺-10 b. CHA-8- H⁺H⁺-1 c. CHA-8- H⁺H⁺-7). Structures of all the other data points are available in the ESI.

9. Figure S8: For 2 Al sites, 32 possible structures should exist. Only 26 have been shown. Plot results for all.

Response: Our manuscript may not have been clear previously, in the section on the enumeration of charge-exchange sites, page 9, we have noted:

“This situation requires a total of 32 calculations per structure. We note that these numbers represent the maximum number of calculations attempted. For some very unfavorable initial positions, the calculations did not converge. This is mostly limited to the GGA level search and happens in less than 10% of the total calculations (for the most part, this is due to placing Pd at the center of a 4MR). The extended number of initial positions in the scheme inevitably leads to some poor initial conditions.”

We also added the following note to Figure S13 on page S16, to further clarify the absence of 6 structures:

“Only 26 of the 32 possibilities are shown due to lack of convergence for 6 structures. These structures failed to converge due to unfavorable initial conditions, as stated in the main manuscript.”

10. Figure 6: Al pairs from the 12 MR are not observed in any results. Since Al pairs with greater connectivity than NNNN were dropped from the study, some of the possible pairs of NNNNNN (diametrical opposites) of the 12 MR are not considered. In a high throughput study, these cases should not be ignored, since the goal of computational studies is to explore all options albeit a strong argument exists not to (e.g., Loewenstein’s rule for not considering N. Even though exceptions exist, this can be considered).

Response: Several Al-Al pairs (13 to be precise) were considered in the calculations. They were grouped with the “other Al pairs” in the plots. We have given those structures a different color (magenta) to clarify that these calculations were considered.

Figure 6: Formation energy of Pd+2 on BEA. Each bar represents unique Al locations. Color coding refers to the type of Al pairs in the zeolite matrix. Patterned bars refer to Al pairs in NNNN positions while solid bars refer to Al pairs in NNN positions. The four most stable sites are illustrated in Figure 5.

Figure 7: Figure 6: Formation energy of Pd+/Pd+H+ on BEA. Each bar represents unique Al location(s). Color coding refers to the type of Al pairs or isolated Al in the zeolite matrix. In Al pairs, patterned bars refer to Al pairs in NNNN positions and solid colors

11. Figure 9: Data shows that the outliers are significant, and comparisons or trends may be studied but conclusions on each category should not be drawn based on this plot. Legend needs to be clear as well.

Response: We agree with the reviewer that in the case of Pd^{2+} we cannot claim a linear relationship. We have clarified this point on page 18 of the revised manuscript:

“There appears to be a linear correlation with a low mean absolute error (MAE) for NO adsorption on Pd^+H^+ -exchanged CHA (mean absolute error is 0.1 eV); however, this is not the case for Pd^{2+} -exchanged CHA.”

In the legend for Figure 8, we have also noted that the lines were drawn only for visual illustration, and we indicated the meaning for half-filled and filled symbols in order to clarify what each symbol represents:

Figure 8: Binding energy versus Pd formation energy on CHA. The color coding refers to the type of Al in then-MR. The marker shapes (\blacklozenge and \diamond) represent Pd^{2+} and $\text{Pd}^+/\text{Pd}^+\text{H}^+$, respectively. Filled markers are used for Al pairs in NNNN configuration and half-filled markers are for isolated Al or Al in NNN configurations. For visual clarity, linear lines based on fitting the data are added (black and blue for Pd^{2+} and $\text{Pd}^+/\text{Pd}^+\text{H}^+$, respectively).

12. Conclusion: Not all Al pair arrangements are considered.

Response: We have clarified in the Conclusion that only Al pairs in 3N and 4N configurations were included in this work, as we found those sites to be the most favorable cation-exchanged sites. The following text was added to page 20:

“We developed a high throughput screening approach for enumerating unique Al (3N, 4N and 5N configurations) arrangements in zeolites,”

13. Figure S11: Legend for squares and circles.

Response: We have added the legend for Figure S16, as requested by the reviewer.

“Figure S16: ΔE_{form} as a function of the distance between Pd and H in Pd^+H^+ optimized geometry on CHA (♦) and BEA (o)”

14. Figure S13: R-squared values need to be reported for parity plots.

Response: We have added the value R-squared for the parity plot (0.086) (page S21)

15. Other experimental work regarding the stability of Al substitutions in CHA and BEA can be used for validation [4,5].

Response: The references cited by the reviewer discuss the Al pairs distribution in zeolites, but do not discuss Pd cation stability. Al distribution in the zeolite is kinetically driven (ref [41]), hence, comparing the thermodynamic favorability computationally is not the best indicator for favorable Al sites. The objective of our work is to examine where Pd cations could locate, and for each Al or Al pair configuration, to determine what site(s) would be best deployed for PNA. We have recently submitted a combined theoretical and experimental study that compares experimental evidence of NO adsorption on Pd-exchanged CHA. The work is titled “Investigation of the Modes of NO Adsorption in Pd/H-CHA”. The comparison between theory and experiment is not trivial, but we attempt in that work to use our theoretical results to explain experimental observations. Below is a sample of the results.

Figure 11: (left) Theoretical estimates of IR stretching frequencies for NO ($\nu_{\text{NO, est}}$) adsorbed on Pd⁺ and Pd²⁺ sites in Pd/H-CHA vs. calculated temperatures of maximum desorption (T_{max}) from the respective sites. Experimental IR spectra collected at 348K, 373K, 473K, 573K, 673K and 773K are shown as solid black lines. The corresponding TPD profile is shown above the graph. (right) Formation energies for the various Pd⁺ and Pd²⁺ sites as defined by reactions 1 and 2.

16. Cite relevant work on QM/MM model from all sources and not just past work from the research group: [6, 7, 8]

Response: We added citations from multiple research groups on applying QM/MM method to study zeolites (page 21)

The QM/MM approach has proven to account for long-range Coulombic and dispersive interactions, which are critical in describing the zeolite framework interactions with adsorbates [40] and has been widely used to model zeolite catalysis/adsorption [32, 37-39, 45-55].

17. Other minor comments: a. Page 4 – Add GGA abbreviation.

Response: We added the following words to explain GGA: “Generalized gradient approximation (GGA) ...” in page 4

b. The data for all generated structures and optimized geometries are missing from the SI.

Response: We have attached all optimized geometries (xyz coordinates) for generated data.

c. Legends of graphs need to be inclusive and there should be no need to read captions to read legends of a plot. (solid color vs crossed color, shape of points, etc.).

Response: We agree with the reviewer’s recommendation and have revised legends for all plots in the manuscript in order to make them inclusive and not require reading the captions.

d. Abbreviation of UV-V on page 18.

Response: “UV-Vis description (ultraviolet–visible)” has been added (page 17)

e. Grammar – end captions with periods.

Response: We have made this correction to all figure captions.

f. Contrasting colors might make plots more legible. Gray color might be changed for a different color like cyan, orange, etc.

Response: We have revisited the coloring used and changed gray to cyan, and also included magenta for 12MR (which we find more contrasting to red compared to orange)

g. Acknowledgement: ...like to thank Saudi Aramco...

Response: We have corrected this statement in the manuscript.

Reviewer #2:

Comments: The authors present a workflow to identify favorable cation-exchanged sites in zeolites. They systematically investigate 696-T CHA and 810-T BEA cluster models with different arrangements of Al pairs. They use QM/MM calculations to identify favorable Pd-exchange sites, and their corresponding influence on NO adsorption performance. However, the novelty of this manuscript is very limited because the whole workflow is simply a combination of conventional approaches.

Response: We agree that our workflow integrates existing components, but we feel that the overall result is novel and transformative: for example, as discussed in the reply to Reviewer 1, we have performed more than 10 times as many QM-based explorations of viable sites as any previous work. Our workflow contains methodology to generate unique Al pairs, automated selection of the QM/MM atoms, scanning potential adsorption sites for the extra-framework cation, performing a layered QM/MM calculations to develop the most favorable location for the extra-framework cation. By building this robust workflow, we were able to determine those new active sites that were not reported in the literature. To the best of our knowledge, there is no similar work pertaining to zeolites that provides such a systematic approach to determining the extra-framework cation location. This was a major driver for us to conduct this work given that the performance of the resulting doped zeolites is directly a consequence of the location of the extra-framework cations.

The authors claim this methodology is easily applicable and transferable to any adsorption/catalysis problem with any zeolite. This is not true. This workflow requires comprehensive computational resources, and it cannot enumerate all possible ways of Al incorporation and Pd-exchanging periodic models, which are closer to real zeolites than cluster models. To reduce the computational costs, the authors significantly decrease the Al contents in these models, which are far away from real zeolites. Therefore, this approach is only applicable to zeolite structures with unrealistically low Al contents. I would recommend rejection as the novelty of this manuscript does not meet the high standard of Nat. Commun. Detailed comments include:

Response: We respectfully disagree with the reviewer that our work lacks novelty. If our approach was apparent and easy, it would already have been done by others. However, our approach enumerates all possible Al pairs, and this applies seamlessly to both cluster systems and periodic systems. To aid other applications, as well as to better explain our approach, we have added further explanation on how this is done (e.g. Figure S3). We have also addressed the concern raised by the reviewer about how the cluster models represent real zeolites later in this response. This is done by showing how cluster models reproduce experimental results with acceptable accuracy based on multiple literature reports and by demonstrating how they compare favorably to periodic calculations, also based on previous literature findings.

DFT calculations remain computationally expensive, despite the recent advances in CPU architecture and the abundance of high-performance computing. We greatly reduced the computational cost by moving from periodic calculations (e.g., BEA unit cell has 192 atoms) to a layered QM/MM approach (average number of atoms is around 50), where we used calculations at GGA level to screen candidates before using the more accurate hGGA calculations. Given that

most of the sites we studied in this work were not reported in the literature, we had to perform the DFT calculation to determine the energy.

We address the concern raised by the reviewer on Si/Al ratio in the next point.

1. The reported Si/Al ratio in Pd-exchanged SSZ-13 was 8.9 (ref. 28), which corresponds to 3.6 Al atoms per CHA unit cell (36T). This manuscript only considers 2 Al atoms in a 696-T CHA cluster model, neglecting a large number of possible Al arrangements. It has been reported in literature that the variation in the content of heteroatoms in zeolite frameworks significantly influence the distribution of extra-framework cations. So, 2 Al atoms in a 696-T cluster cannot represent the real situation, where 3.6 Al atoms are distributed in a 36-T unit cell.

Response: In the cluster model employed in our work, the active site is modeled using a high level of theory (QM), and the remaining part of the cluster is modeled using a standard molecular mechanics (MM) approach. The objective of the MM region is to capture the long-range interaction (as detailed in the reference by E. Mansoor). In the example of CHA, we used 696 T-atoms, and only included 2 Al in the cluster model. However, this does not mean the Si/Al ratio is 696/2. In addition, we added to the following to the manuscript to address the accuracy of the QM/MM approach, and how modeling only the active site as the QM region reproduces experimental observables (page 21):

The QM/MM approach has proven to account for long-range Coulombic and dispersive interactions, which are critical in describing the zeolite framework interactions with adsorbates [40] and has been widely used to model zeolite catalysis/adsorption [32, 37-39, 45-55]. Many studies have shown that the QM/MM approach gives a good prediction of experimental data for different zeolites and adsorbates [17, 32, 37-40, 49-51, 53-55]. This approach is also computationally more efficient than periodic calculations since it requires a smaller number of QM atoms, especially when the unit cell contains a large number of atoms (e.g. BEA and FAU). A number of studies benchmarked QM/MM calculations against periodic calculations and found similar results [31,45].

2. The poor data in Figure S7 is not sufficient to support the authors' claim that intermediate O-O distance yields more favorable arrangements.

Response: We have revised the plot and added 4 additional plots for other zeolite structures. In each plot, insert image resembling structures at short, intermediate, and long O-O distances are shown. We have also added to the supporting information all structures studied in this work, and their respective energies. The structures are shown earlier in this response (Figures S8-S12).

3. Please provide structure illustrations of all the structures mentioned in this manuscript, such as BEA-36, BEA-59, and BEA-62.

Response: We have added the images of these structures (and all other structures explicitly mentioned in the manuscript) to the supporting information. Detailed xyz coordinates of all other structures studied in this work are available in the electronic supporting information.

4. Molecular dynamic method is recommended for NO adsorption calculations.

Response: There are many merits to molecular dynamics calculations. However, we note that adsorption is thermodynamically driven. We wanted to study the thermodynamic driving force for cation placement, and NO adsorption on those cation-exchanged sites. DFT is better suited for this purpose, since it considers electron-electron interaction, which MD does not consider (unless it is ab-initio MD, in which case the calculations would be far too expensive). It is virtually impossible to do ab-initio MD calculations on the same number of structures we did in this work (>7,000 explicit DFT calculation).

5. Some unfavorable charge-exchanged sites (e.g. Pd⁺ or H⁺ at 4 or 5 MR) that are inaccessible in the NO adsorption process can be excluded before QM/MM calculation to reduce computational cost.

Response: We wanted to first address Pd cation locations in the zeolite structure, irrespective of the application (other applications for example could have a smaller guest molecule interacting with the Pd cation). In NO calculations on CHA, for all Al configurations, the most favorable Pd cation configuration is always accessible to the NO molecule (the open cage site), as shown in the optimized geometries in the electronic supporting information. Hence, the suggestion made by the reviewer does not result in further reduction of computational cost for CHA. However, this certainly applies to many other zeolites.

Reviewer #3:

1) There have been several computational studies on the distribution of cations in Al-containing zeolites, such as those of carried out by Sauer, Dedecek, Sklenak, Nachtigall, Newsam, Mellot, Cheetham, Ruiz-Salvador, Sastre, Catlow, etc. These studies have provided a large body of knowledge in the field, which has not been discussed neither in the introduction nor in the discussion. The present study gives a feeling of isolation, since the results are not put into the context of the state of the art; the authors should present an analysis of the relation between the present results and relevant results present in the literature, mentioning how their results agree or disagree with previous results.

Response: We agree that the previous version of the manuscript did not include sufficient citations to the large body in the literature that contributed to the study of zeolites. We have now included citations from Sauer, Sastre Catlow, Dedecek and many other research groups (60 total citations) in the revised version of our work. Since the objective of this work is to overcome limitations of theoretical work in determining the cation-exchanged site, we focused on the literature work pertaining to this topic.

Below are highlights of the changes to the manuscript to address this shortcoming (page 2):.

It is, therefore, evident that the selection of a representative cation site is challenging and that full exploration of the chemistry on all possible sites is computationally formidable because of cost. For these reasons, many higher level theoretical studies of zeolite-catalyzed reactions [9-12] using density functional theory (DFT) have chosen to focus on a few active sites selected on the basis of limited experimental evidence and/or physical intuition. While this choice leads to a more tractable set of calculations, the downside is that potentially important active sites might be missed because they are difficult to identify experimentally or are not physically intuitive. Indeed, growing numbers of DFT studies have progressed from studying single T-sites [13,14] to address greater complexity [15,16].

Comparison between the number of calculations in this work and other literature studies (page 4):

This approach allowed us to calculate over 7,000 explicit DFT calculations in a systematic way, which is more than an order of magnitude higher than what is done typically in DFT studies of zeolites [13, 14, 16, 30-32].

The finding of Al pairs in the 6MR yielding the most stable Pd cation site (page 12):

Al pairs in the 6MR arrangements, especially in the NNNN position, provide the most favorable host for the Pd cation, in which case the Pd resides at the center of the 6MR (Figure 5a and b and Figure S15a and b). This finding is consistent with other recent studies [28,29].

The role of Pd speciation in NO update (page 19):

Previous literature work showed that several factors affect NO adsorption on Pd-exchanged zeolite (e.g. how Pd is introduced to the zeolite support) [45-47]. Isolated Pd cations have been shown to be desired for PNAs, [46,48-50]. The exact nature of the oxidation state of the Pd cations and their location in the zeolite continue to be debated in the scientific literature [28-30]. Experimentally, there are direct evidence of Pd⁺ presence in the zeolites, as shown in the electron paramagnetic resonance spectroscopy (EPR) results [48,51,52]. Although it is unclear if Pd⁺H⁺ is present under operating

conditions; however, Figure 8 indicates that if present, it is a superior NO adsorption site compared to Pd^{+2} . In addition, by enumerating the possible Al pairs in the zeolite framework in this work, we determined the sites that are most critical for NO uptake in the framework.

How the Al pairs in the 6MR, although most extensively studied, does not reflect the optimum adsorption site for NO (page 19):

Figure 8 also shows that Al pairs in the 6MR arrangement (especially in NNNN configuration), which has been discussed most extensively in the literature because they hold Pd^{+2} cations most stably, are weaker sites for NO adsorption compared to other Al arrangements. This is not unexpected since the more stable the Pd, the more weakly it can bind to a guest molecule. It is striking, however, that the difference in ΔE_{NO} compared to the other Al arrangements (0.25-1.5 eV). Figure 8 also highlights how many of the CHA sites are very close in energy, especially for $\text{Pd}^{+}\text{H}^{+}$. Therefore, it is very difficult based on the small differences in energy values to discern spectroscopic data and assign them to specific sites. The results indicate that an ensemble of sites of very similar energies contribute similarly to the adsorption of NO.

Accuracy of the QM/MM approach (page 21):

The QM/MM approach has proven to account for long-range Coulombic and dispersive interactions, which are critical in describing the zeolite framework interactions with adsorbates [40] and has been widely used to model zeolite catalysis/adsorption [32, 37-39, 45-55]. Many studies have shown that the QM/MM approach gives a good prediction of experimental data for different zeolites and adsorbates [17, 32, 37-40, 49-51, 53-55]. This approach is also computationally more efficient than periodic calculations since it requires a smaller number of QM atoms, especially when the unit cell contains a large number of atoms (e.g. BEA and FAU). A number of studied benchmarked QM/MM calculations against periodic calculations and found similar results [31,45].

2) The different configurations for the distribution of cations are obtained using an energy-based method, but a more rigorous approach would be to use cation distribution methods based on the symmetry of the crystal, such as those implemented in the programs SOD or Supercell. These methods could help not only to study the distribution of Al atoms, but also in the distribution of extra-framework cations.

Response: We find it essential to use an energy-based method since we are looking for the thermodynamically favorable site. A symmetry-based approach cannot determine which sites are more favorable. Moreover, the symmetry of the zeolite unit cell becomes more complex with the introduction of multiple Al atoms. To the best of our knowledge, both SOD and Supercell do not offer a viable solution to this problem. We have also searched elsewhere and could not find an alternative solution. Given the lack of available symmetry models in the literature and the lack of clarity on whether such model could give the most optimum location for the extra-framework cation, an energy-based approach (such as the one implemented in this work) is likely the most suitable option.

3) The authors use QM/MM models to study the zeolites. This approach has been used due to its computational cost. But recent advances both in hardware and software make feasible the study of

a large number of configurations using purely QM periodic calculations. For instance, I guess that the present study could have been carried out using CP2K, which would have allowed the proper inclusion of periodicity and long-range interactions, and the errors introduced by spurious interactions between the QM and MM regions could have been avoided. It is now too late to do the calculations, but the authors should mention how much they estimate that these errors could be.

Response: Using a QM/MM approach was a major reason we were able to conduct over 7,000 explicit DFT calculations for zeolites (which typically include larger number of atoms). We clarified in the manuscript how QM/MM approach has been widely used in the literature to study zeolites, how it compares well with experimental results (refs. [17, 32, 37-40, 49-51, 53-55]), and how favorably it compares with periodic DFT calculations (ref[31,45]). If the calculations were done using CP2K or QuantumEspresso (another periodic DFT library), the number of potential calculations completed would be reduced by close to a factor of $\frac{1}{4}$, since BEA/CHA structures in periodic systems contain $> 4x$ of the atoms in the QM/MM approach.

Another disadvantage of periodic DFT calculations, as explained in ref. [40]: “The embedded-cluster models are beneficial as they remove the periodic boundary conditions that can hinder charge non-neutral models and/or high-level calculations of large unit cell systems, such as zeolites, whilst maintaining the correct long-range electrostatics for the active site of interest”.

Given the acceptable accuracy of the QM/MM calculations and the efficient use of computational resources, we find the QM/MM approach to be the most suitable for this work.

4) This comment is somehow related to the previous one. Many experimental and theoretical studies have shown that the introduction of extra-framework cations does have a relevant effect in terms of distorting the zeolite structure (in some cases even leading to the collapse of the structure), which in turn might change the positions of the cations themselves. This effect is largely neglected in the present study, since the atoms in the MM region are fixed, so the authors should at least mention how the results are likely to be affected by this neglect.

Response: We address this comment in two ways: First, Lardinois et al. (*Chem. Mater.* 2021, 33, 5, 1698–1713) shows side-by-side XRD spectra of H-CHA and Pd-H-CHA (Figures S2.5 and S2.6). The two XRD results are identical, confirming that the structure is not distorted upon introduction the cation nor does it collapse. Second, we have added the optimized geometries of the four structures studied in Figure S5 (where the number of atoms in QM region is varied) and showed that the atomic positions of those structures do not change upon them being changed from the QM region to the MM region.

5) Finally, the manuscript has several typos and minor language errors that should be corrected.

Response: We have reviewed the manuscript carefully and corrected the few mistakes found.

New results added to the revised manuscript

1. Selectivity of Pd-exchanged CHA in removing NO from other gas species in automobile exhaust:

We added the following information to the revised manuscript (page 19):

“We further extend our work to analyze the selectivity of Pd-exchanged-CHA toward NO compared to other gaseous species present in automobile exhaust (namely CO, CO₂, and H₂O). We limit the calculations to Pd⁺ and Pd⁺H⁺ exchanged CHA based on its stability (Figure 9). The results presented in Figure 10 show that other adsorbates follow a linear scaling relation, similar to that for adsorbed NO (MAE for each species is <0.1 eV). Among the studied gaseous species, NO adsorbs the most strongly to Pd-exchanged-CHA. CO also adsorbs strongly, only 0.1 eV less than NO. Water and CO₂ adsorb much more weakly (on average 0.6 and 1.2 eV less than NO, respectively), showing that Pd-exchanged-CHA is an excellent adsorbent at selectively removing NO from the other species in automobile exhaust. We also observe that as the Pd formation energy decreases, the gap in adsorption energy between NO and the other species becomes more pronounced, likely a result of the Pd electron density being mostly used in binding to the zeolite framework.”

Figure 9: Binding energy of a number of gaseous species on Pd-exchanged CHA (Pd⁺/Pd⁺H⁺) against Pd formation energy (at unique Al configurations). Dotted lines refer to fitted data for each gas adsorbate. Detailed structures for each data point are available in the SI

2. Comparison between Pd⁺², Co⁺² and Ca⁺²:

We added the following text to the revised manuscript (page:17):

“Given that the energetic order of Al configurations in BEA does not correlate directly between Pd +1 and +2 oxidation states, we also explored whether or not the Pd⁺² results correlate better with other divalent metal cations (Co⁺² and Ca⁺²). We performed a limited number of calculations on the other cations on CHA and found some correlation with Pd⁺² results at more negative formation energies, but no consistent trend at less negative formation energies (Figure S20). This indicates that the results for one cation in a zeolite matrix are not readily transferable to other, even if chemically similar cations exchanged into the same zeolite. This finding is important, since the ultraviolet-visible (UV-Vis) spectrum of Co⁺² is sometimes used to identify the location of divalent cations in zeolites [42, 43].

Figure S20: Comparison between favorable Al pair configurations for Pd⁺², Co⁺², and Ca⁺² exchanged CHA.

3. Al-Al pairs in 5N configuration

In our previous submission, we assumed that at the limit of infinite separation between Al pairs, each Al would behave as an isolated Al. We have assumed that this starts taking place at 4N (where the two Al pairs are separated by two Si atoms). Per the recommendation of the first reviewer, we revisited and tested this assumption by doing the calculations on 5 different CHA structures where Al pairs are in a 5N configurations. For each unique Al pair, a search for the most stable cation position was done. We found that in all 5 cases, the Pd⁺² formation energy of each site is weaker than any of the studied Al pairs in this work (both 3N and 4N configuration). We added the following to the manuscript (page 6):

“Pairs involving Al atoms farther apart than an NNNN configuration were not considered since as the Al-Al distance increases, the impact of having the Al pair diminishes and each atom starts behaving as an isolated Al site (some tests demonstrating this point are shown in Figures S1 and S2)”

Figure S1: Images of the optimized geometry of 5 different Pd²⁺-exchanged-CHA structures where the Al pairs are in a 5N configuration. The same color coding as in Figure 5, however, for clarity purposes, Al atoms are enlarged and are shown in green. The calculated formation energies of the structures are a. -0.72 b. -0.72 c. -0.84 d. -1.05 and e. -1.22 eV.

Figure S2: Pd²⁺-exchanged-CHA formation energy as a function of Al-Al distance. Color coding refers to the type of Al pairs.

4. Al pairs on the opposite side of the MR and Al pairs in 12 MR

In our previous submission, we did not consider Al pairs where the atoms are on opposite side of the 12MR in BEA. In preparing this revised version, we have revisited this assumption by doing the calculations on all those structures (14 in total) and have included their data in Figures 6 and 7. We have also added color coding to Al pairs in the 12 MR to distinguish them from the other Al pairs. The most stable sites, however, remain the same as was presented earlier. Please note that the numbering of structures has changed (all structures are available in the SI).

Figure 6: Formation energy of Pd+2 on BEA. Each bar represents unique Al locations. Color coding refers to the type of Al pairs in the zeolite matrix. Patterned bars refer to Al pairs in NNNN positions while solid bars refer to Al pairs in NNN positions. The four most stable sites are illustrated in Figure 5.

Figure 7: Figure 6: Formation energy of Pd+/Pd+H+ on BEA. Each bar represents unique Al location(s). Color coding refers to the type of Al pairs or isolated Al in the zeolite matrix. In Al pairs, patterned bars refer to Al pairs in NNNN positions and solid colors refer to Al pairs in NNN positions. The most stable site (BEA-78) is illustrated in Figure 5.

REVIEWERS' COMMENTS

Reviewer #1 (Remarks to the Author):

Accept

Reviewer #3 (Remarks to the Author):

The manuscript has been improved compared to the previous version, but I still find it unsuitable for publication in Nature Communications, for two main reasons. On the one hand, the workflow does not seem to be significantly different from what it is commonly performed when carrying out QM/MM studies, it is mainly a series of steps, going from low to high levels of theory, with which to find a set of possible structures. I do not see a significant breakthrough in the method for creating and selecting the structures, it still has a "brute force" feel. On the other hand, the results obtained with this method are not very relevant. They have found some Al configurations for Pd+H⁺ in which Pd cations bind very stably, as well as some unexpected stable locations for Pd. They have also found that Al pairs in 6MR are very stable sites for the Pd cation, but NO adsorption there is weaker than in other sites. I think that these results will not be of interest for the wide readership of a journal such as Nature Communications, and therefore I do not recommend its publication in this journal, although it is a very good article that would fit in a more specialized journal.

Response to reviewer comments (second round)

Reviewer #1 (Remarks to the Author):

Accept

Reviewer #3 (Remarks to the Author):

The manuscript has been improved compared to the previous version, but I still find it unsuitable for publication in Nature Communications, for two main reasons. On the one hand, the workflow does not seem to be significantly different from what it is commonly performed when carrying out QM/MM studies, it is mainly a series of steps, going from low to high levels of theory, with which to find a set of possible structures. I do not see a significant breakthrough in the method for creating and selecting the structures, it still has a “brute force” feel. On the other hand, the results obtained with this method are not very relevant. They have found some Al configurations for Pd+H⁺ in which Pd cations bind very stably, as well as some unexpected stable locations for Pd. They have also found that Al pairs in 6MR are very stable sites for the Pd cation, but NO adsorption there is weaker than in other sites. I think that these results will not be of interest for the wide readership of a journal such as Nature Communications, and therefore I do not recommend its publication in this journal, although it is a very good article that would fit in a more specialized journal.

Response: We appreciate that the reviewer feels our revised paper is improved. We reiterate the fact that we addressed the main issues raised previously by the reviewer, such as by demonstrating the ability of our high throughput workflow to locate previously unidentified stable placements of Pd ions in CHA and BEA, and also showing selectivity of Pd-exchanged CHA to NO adsorption vs other exhaust gases.